# Trajectories toward maximum power and inequality in resource distribution networks

**Natalie Davis**[1,2]*, **Andrew Jarvis**[1], **M. J. Aitkenhead**[2], **J. Gareth Polhill**[2]

**1** Lancaster Environment Centre, Lancaster University, Lancaster, United Kingdom, **2** The James Hutton Institute, Aberdeen, United Kingdom

* ndavis.research@gmail.com

## Abstract

Resource distribution networks are the infrastructure facilitating the flow of resources in both biotic and abiotic systems. Both theoretical and empirical arguments have proposed that physical systems self-organise to maximise power production, but how this trajectory is related to network development, especially regarding the heterogeneity of resource distribution in explicitly spatial networks, is less understood. Quantifying the heterogeneity of resource distribution is necessary for understanding how phenomena such as economic inequality or energetic niches emerge across socio-ecological and environmental systems. Although qualitative discussions have been put forward on this topic, to date there has not been a quantitative analysis of the relationship between network development, maximum power, and inequality. This paper introduces a theoretical framework and applies it to simulate the power consumption and inequality in generalised, spatially explicit resource distribution networks. The networks illustrate how increasing resource flows amplify inequality in power consumption at network end points, due to the spatial heterogeneity of the distribution architecture. As increasing resource flows and the development of hierarchical branching can both be strategies for increasing power consumption, this raises important questions about the different outcomes of heterogeneous distribution in natural versus human-engineered networks, and how to prioritise equity of distribution in the latter.

## Introduction

Both biotic and abiotic systems require energy for maintenance and growth, necessitating the relocation of energetic resources from points of supply to points of consumption and end use. This need for energy drives the development of resource acquisition, distribution, and end-use (RADE) networks [1] in all earth systems. RADE networks are by definition spatial structures, constructed with both physical materials, such as asphalt, wire, or connective tissue, and informational cues, such as scent trails or memories. Additionally, all RADE networks can be conceptualised as a collection of resources, where the energy flow is generated and supplied; end-use consumers, where the energy flow is required; and the links between them. The construction, maintenance, and use of these networks inevitably requires a considerable proportion of the resources available to consumers. As it is evolutionarily advantageous to maximise the net

**Data Availability Statement:** All relevant data are within the manuscript and its Supporting Information files.

**Funding:** ND acknowledges funding from a joint Lancaster University/James Hutton Institute PhD

studentship. JGP acknowledges funding from the Scottish Government Rural Affairs, Food and Environment Strategic Research Programme 2016–2021 (https://www2.gov.scot/Topics/Research/About/EBAR), Work Packages 2.4 "Rural Industries" and 3.3 "Food Security." The funders had no role in study design, data collection and analysis, decision to publish, or preparation of the manuscript.

**Competing interests:** The authors have declared that no competing interests exist.

resources available for further growth and development [2,3], there is significant adaptive pressure to drive RADE network development toward increasing efficiency. Additionally, these networks often share common forms such as hierarchical branching, and serve end consumers operating in highly heterogeneous states. Rarely, if ever, are these two observations explicitly associated, but given the role of RADE networks in determining the states of the consumers they support, correlation between network topology and variance in supply to these points of end use should be expected. Establishing this connection is crucial, not only in natural systems as a means of accounting for variability, but especially in social systems where inequality is of such profound importance.

Inequality in human society is typically conceived as an outcome of combined social, political, psychological, and economic influences. Although many theories about the origins of inequality include discussion of resources, such as their economic defensibility, most theories still invoke cultural or technological arguments as well [4]. Additionally, even arguments based on instincts and social behaviour rarely connect these to resource distribution explicitly [5], despite the essential role of resource movement in giving rise to any cultural, technological, and social forces. This gives the appearance of resource distribution and emergent inequality in social systems as having fundamentally different causes than hierarchies in environmental and biological systems, or energetic niches in ecosystems. Moreover, while energy consumption is not typically the named objective of economic management, the drive toward ever-increasing economic growth still requires energetic resources to build and maintain the infrastructure that generates returns [6], paralleling the energy used for growth and maintenance within natural systems. As both natural and human-engineered systems rely on resource distribution networks to relocate energetic resources, it seems logical to consider heterogeneity within the networks and resources themselves as potentially foundational causes of inequality [7]. However, a formal, quantitative linkage between RADE network architecture, inequality in resource distribution, and the rate of increase of that inequality during network development, has not yet been elucidated.

RADE networks are theorised to develop in a way that maximises the availability of resources to points of end use, such that these end consumers capture the maximum free energy for their own purposes in doing 'useful work' [2], such as increases in growth, development, or storage [8]. This is formalised in the Maximum Power Principle (MPP), which states that, given adequate degrees of freedom, a system will self-organise so as to maximise its power output, or capture and use of free energy per unit time [9]. An explanation for why such behaviour would emerge is that increasing the availability of useful energy currently within a system allows the system to capture more free energy in the future, such that MPP is simply the expression of a growth-orientated positive feedback, which inevitably evolves to some boundary or constraint. Often these constraints can be considered thermodynamic limits on efficiency [10]. Hereon, this maximisation of energy consumption and power production will be referred to as 'maximum power,' to include the transfer or capture of free energy, and its consumption in performing useful work. MPP is closely related or equivalent in many systems to the Maximum Entropy Production Principle (MEPP) and related thermodynamic extremisation principles (see e.g. [11,12]). While criticisms of both MPP and MEPP [13–15] have been put forward, these have mostly been resolved through clarification and restrictions to the theories [16,17]. As such, these extremisation principles provide a framework and directionality for evolution and systems progression, and can be used to help understand broader trajectories for systems development, and network development within that [12].

Specifically, systems often maximise power via changing state with respect to available energy inputs and constraints; changing network architecture to take advantage of untapped resources or minimise energy consumption in transporting resources; or both. Some theorise

that the development of self-similar hierarchical branched networks, seen in a diverse array of naturally-occurring and human-engineered systems, including vascular networks in plants and animals, power grids, and river basins, is an example of the latter strategy [18,19]. Resource flows transmit energy using a mass carrier, such as food or electrons; and during transmission these carriers experience frictional dissipation when moving over distances. This creates the evolutionary pressure to minimise transmission distance to maximise the energy transferred, hence the development of optimal space-filling structures such as hierarchical branching. Despite the theoretical universal drive toward increasing levels of energy consumption, there has been limited study on the relationship between this increasing trajectory, the architectures favourable to it, and the impact that has on the inequality of energy distribution in ecological and socio-ecological systems, as introduced above.

Since frictional dissipation derives from distance, spatially explicit modelling of RADE networks is crucial to understanding their development and dynamics, and the impacts these have on inequality. The dynamics of energy-mass flows over distances are described by a group of phenomenological linear flow laws, including Ohm's law for electrical current, Darcy's law for fluid flow, Fick's law for diffusion, and Fourier's law for heat transport. These flow laws state how force and flux are closely related to one another [20], making them useful for modelling a diverse range of energy-mass flow systems. It is hypothesised that, when viewed from the appropriate perspective, physical systems such as ecosystems and socio-ecological systems should all follow these force-flux relationships [21]. Odum in particular made extensive use of electrical analogue modelling, which calculated the flows through a system using Ohm's law, by identifying the analogous concepts to voltage, current, and resistance or conductance in a system (see e.g. [22,23]). While his focus was on interactional models such as food webs, less work has been done applying this type of modelling to spatially-explicit networks, where the friction or resistance term, or equivalently the latter's inverse, conductance, is related to the physical distance the flows must cover (although see specific case studies in [24,25]). Drawing analogies between resource flows in complex coupled socio-ecological systems and electrical circuits can be criticized because the formulas underlying analysis of electrical systems are linear, while those of the former are nonlinear. However, Wang et al. (2012) argue that many systems show linear behaviour at macroscales or microscales, and these can be modelled individually and recombined. Such linear models thus remain useful analogies for exploring generalized realistic systems [26], and may still result in the emergence of complex properties. Exploration of the effects of nonlinear formulas on these observations is the potential subject of future work.

Given the theoretical argument and empirical evidence for systems to evolve toward a state of maximum power, this paper will explore the potential relationships between the trajectory towards maximum power, RADE network structure, and inequality. It will thereby generate further insight into the characteristics of complex spatially explicit RADE networks as they develop toward and operate at maximum power. Specifically, systems will be modelled with representative electrical circuits to elucidate the dynamics and characteristics of generalised RADE networks evolving toward maximum power transfer, explore characteristics of those networks and the evolutionary levers employed in their development, and discuss how these relate to the existence and development of inequality between end consumers in those networks.

## Inequality as a function of network architecture and resource flows

### Modelling framework using an electrical analogue

In mass-flow networks, the flow through the network is generally conceptualised as a function of the driving potential gradient, and the characteristics of the material through which it flows.

As introduced above, this relationship can be represented in a given system using an analogue of one of the phenomenological flow laws, such as Ohm's law,

$$I = \frac{\Delta V}{R}. \tag{1}$$

In the framework here, $\Delta V$ is the potential gradient driving the flow between two points in the network, $I$ is the resource flow, and $R$ is the resistance of the associated link, a measure of the friction encountered by the flow, given by the ratio of link length to link strength or capacity, $R = \frac{L}{S}$. The power output $P$ delivered to a given end-point consumer $c_i$ in the network, or final power, is defined as

$$P_{C_i} = I_{C_i} V_{C_i}. \tag{2}$$

Alternatively, $\Delta V$ can be conceptualised as the energy consumed in transport, whether active or passive, as the power consumed in transport between two points, $P_L$, is given by combining Eqs 1 and 2 as

$$P_L = I\Delta V = I^2 R. \tag{3}$$

The relationship between this power consumption in transport and the spatially-related resistance term clarifies the evolutionary pressure for a system to minimise resistance, such as through the development of increasingly efficient structures that are hypothesised to minimise frictional losses [18]. Specifically, minimising the frictional losses maximises the rate of energy transfer, or power, at the spatially disparate points of final dissipation or consumption.

Along with reorganisation of network architecture to minimise resistance, systems can evolve toward higher final power by adapting network state with respect to the quantity and potential of available resources. For example, the increased availability of resources in summer months allows mammals to operate at a higher metabolism and in a greater geographic range, whereas hibernation is an adaptation to decreased resource availability in the same range during winter months [27]. In the framework here, adaptation of network state can be represented by changing $I$, or by changing the potentials that comprise $V_C$. In the former case, increasing $I$ causes $P_{C_i}$ to increase (Eq 2), until the increased frictional losses from higher resource flow (Eq 3) causes a large enough increase in $\Delta V$, such that $P_{C_i}$ decreases. In this way, the trade-off between $I$ and $\Delta V$ is mediated by $R$, again providing evolutionary pressure for a system to develop lower resistance, as it increases resource flows.

Due to this trade-off between $I$ and $\Delta V$, maximum final power occurs in this framework when the potential at the consumer is half the potential at the resource (see S1 Text for derivation). This is consistent with the Maximum Power Transfer theorem for electrical circuits [28], empirical findings of maximum power in natural systems such as streamflow [29], muscle contraction [30], sediment transport [11], and the Maximum Power Principle as extended to generalised interacting components [9]. In electrical circuits, simplification algorithms such as Thévenin's theorem [28] allow for complex circuits to be represented by simpler equivalents. Similarly, in the framework presented here, the relationship between consumer and resource potential can be extended over the entire network using the network mean values for power consumption, resource flow, resistance, and potentials. Specifically, the network-wide maximum final power state is then

$$\overline{V_C} = \frac{\overline{V_R}}{2}, \tag{4}$$

where $\overline{V_C}$ and $\overline{V_R}$ are the network mean values for consumer and resource potential, respectively.

In order to extend this framework to explore the heterogeneity among consumers within the network, and how this is affected by increasing consumption and changing network organisation, the relationship between consumer potential, resource flow, and resistance can also be expressed in terms of the respective standard deviations. Although it is more common to use the Gini coefficient or other relative measures to quantify inequality in economic and similar analyses, these can obfuscate increases in absolute inequality when the relationship between variables stays constant [31,32]. For example, if each number in a distribution is increased by 50%, the standard deviation of the distribution increases by 50% and the range by 67%, but the Gini coefficient remains the same as the relative relationships are unchanged. Moreover, the Gini coefficient and similar metrics are unitless measures, whereas the standard deviation has the same units as the mean. Any relationships elucidated involving standard deviation will therefore be more consistent with those identified above using means.

The distributions of consumer potentials and final power consumption in a network are the result of the spatial distribution of consumer and resource nodes and links, and the magnitude of resource flow. In networks where there is a single direct connection from each consumer to a resource point, equal resource flow to all consumers, and no interconnections between consumers, the standard deviation of the consumer potentials is $\sigma_{V_C} = \sigma_R I$, derived from Eq 1, and the standard deviation of consumer final power is $\sigma_{P_C} = \sigma_R I^2$. Therefore, in networks with equal resistances along all links, such as an idealised radial burst network, the standard deviations of consumer potentials and final power consumption would be zero. In contrast, increasing resource flows along links with unequal resistance would cause an increase in the standard deviations of potential and final power consumption, due to unequal decreases in consumer potentials.

In more interconnected networks, however, the standard deviations of consumer potential and final power consumption are complex properties, as changes in potential at one node would propagate to interconnected nodes throughout the entire network. As such, determining the baseline structural heterogeneity of the network helps isolate the effects of spatial distribution and connectivity from those of resource flow in increasing the distributions of consumer potential and final power consumption. Here, the 'effective resistance' $R_{E_i}$ is the resource flow-normalised drops in potential from a resource to a given consumer $i$,

$$R_{E_i} = \frac{V_R - V_{C_i}}{I_i}. \qquad (5)$$

As opposed to the traditional measure of resistance, which is calculated for a given link, the effective resistance is calculated along the whole path between a given consumer and resource, even if the two nodes are connected indirectly via multiple links. The effective resistance therefore considers the interaction effects along the links, as well as the real resistances of the link or links between a consumer and resource: its standard deviation relates the heterogeneity in physical distances around the network that the flows cross, network connectivity, and the quantity of flow, to the disparities in consumer potential or power.

In the special case of direct connections between consumers and a resource, the effective resistance simplifies to the link resistance. In all networks, therefore, the standard deviation of effective resistance is the constant of proportionality between the standard deviation of consumer potential or power, and resource flow, such that $\sigma_{V_C} = \sigma_{R_E} I$, and $\sigma_{P_C} = \sigma_{R_E} I^2$. As with the traditional measure of resistance, effective resistance and its standard deviation are stationary for any quantity of resource flow through a given link in the network architecture. It is clearly influenced by the connectivity and symmetry of the network, as asymmetry in path

length, Euclidean distance, or number of intermediary or downstream nodes all increase the inequality in consumer potential and final power consumption. Notably, since effective resistance includes the effects of both physical structure and connectivity, it could potentially be a useful mapping between spatial and relational dimensions of networks, which have typically been analysed separately.

## Simulations to illustrate framework

To illustrate these described dynamics of resource flow in networks, generalised RADE networks were modelled using the relationships presented above. Initially, the networks were comprised of only two types of nodes distributed in space: resource supply nodes and consumer nodes. Consumer nodes could be connected to one another, such that the consumers who were more directly connected to resource nodes passed resource flow along the network to more distant consumers. However, this was limited to the excess resource flow remaining after the initial consumers had met their requirement: consumer nodes could not act as resources to generate additional flow. The resistance was held constant across all links, and was modelled as the ratio of link length to strength, as described previously. The networks were evolved toward maximum power by increasing the resource flows through them and determining the distribution of power consumption across the network using a matrix inversion. The full details are provided in Section 5. This approach, modified from load flow analysis in electrical grids [33], ensured that the resource flows calculated for each node were consistent with the constraints of the first and second laws of thermodynamics, as resource flow was conserved, and power losses around the network were proportional to the size of the network. A sample of the networks simulated is shown in Fig 1, and complete results are in S1 Table.

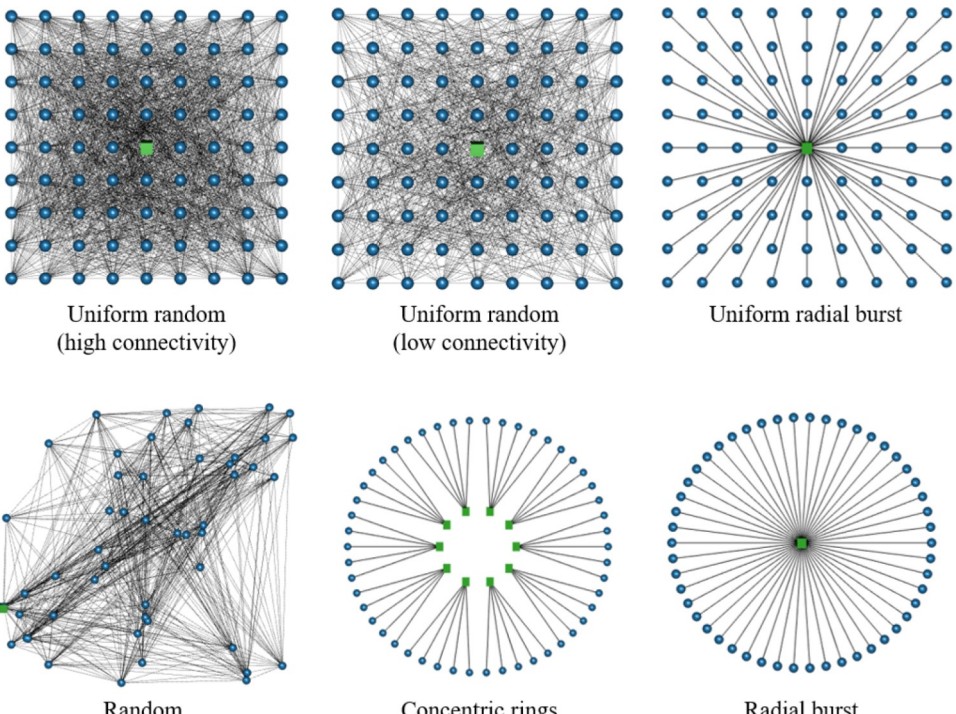

**Fig 1. A sample of the networks used to simulate evolution toward maximum power.** The green squares are resource nodes, and the blue circles are consumer nodes. The grey lines are links between them. Maximum power was calculated by varying the resource flow through the network and calculating the total final power across all consumer nodes.

The outcomes of a representative sample of the simulations are shown in Fig 2. As consistent with Eq 4 above, in all simulations maximum power occurs when the mean consumer operates at 50% of the potential of the mean resource (Fig 2A). Moreover, the relationship between resource flow per consumer squared, $I_c^2$, and the standard deviation of consumer power, $\sigma_{P_C}$, is linear (Fig 2B), with slope $\sigma_{R_E}$, as calculated by least-squares regression and plotted against the estimate using Eq 5 (Fig 2C). This heterogeneity of distances and connections between the consumers and resource causes a distribution of consumer potentials, reflected in $\sigma_{P_C}$. The relationship between consumer potential and power heterogeneity for the different networks, and the resource flow per consumer, is also shown in Fig 3, where increasing $I_C$ over the course of the simulation, and hence decreasing $\overline{V_C}/\overline{V_R}$, causes $\sigma_{P_C}$ to increase.

The networks that show more heterogeneity in the Euclidean distance, path distance, or both, and less connectivity between consumers, have higher inequality as measured by $\sigma_{R_E}$ (Figs 1 and 2C). This suggests that connectivity among consumers can also play a role in limiting the inequality in frictional losses and the resultant consumer potential heterogeneity. This mechanism is perhaps similar to the translocation of nutrients through fungi, where symbiotic connections between the mycelium and plant root systems allow for the redistribution of heterogeneously-located nutrients, providing more remote portions of the mycelial network greater access to resources [34].

While the resource nodes in these simulations operated at a constant potential, similarly to time-averaged behaviour of renewable resources, or a system observed over a short timeframe, these results suggest that inequality would increase even more quickly in systems with diminishing resources. This would be because the less optimally located and connected consumers would experience larger decreases in power, due to the decreasing resource potential amplifying the effects of their higher effective resistance. This is a current line of investigation for an extension of this work.

## Inequality in branching networks

### Branching as a strategy to increase the maximum power of a system

Although changes in state variables, such as potential, allow any given network architecture to achieve its maximum power, this maximum can be increased further through the evolution of the network architecture itself, as discussed. In the framework presented here, this would be illustrated by network reorganisation or otherwise reducing $R$, such that higher resource flows do not cause as much frictional loss (Eq 3). This does not necessarily require decreasing $\sigma_{R_E}$ however, as theoretically the distribution of effective resistances could remain the same for a different configuration of actual resistances.

One means by which systems evolve toward increased consumption through network change is through self-organisation into hierarchical branching structures, which are prevalent in both naturally-occurring and human-engineered systems [18,19]. In these networks, multiple downstream consumers may draw resource flow from the same resource, although this causes increased frictional losses by increasing the $I$ term in Eq 3. This is offset in many systems by the development of higher-capacity links along shared pathways, such as preferential flow paths [11,35]. This is equivalent to varying the link strength in the equation for $R$ (see Methods).

### Branching simulations

To illustrate the dynamics of branching networks more clearly, another set of simulations was performed, featuring idealised self-similar hierarchical branching networks. In these simulations, two networks were constructed. In the first, the network had consumers arranged in a

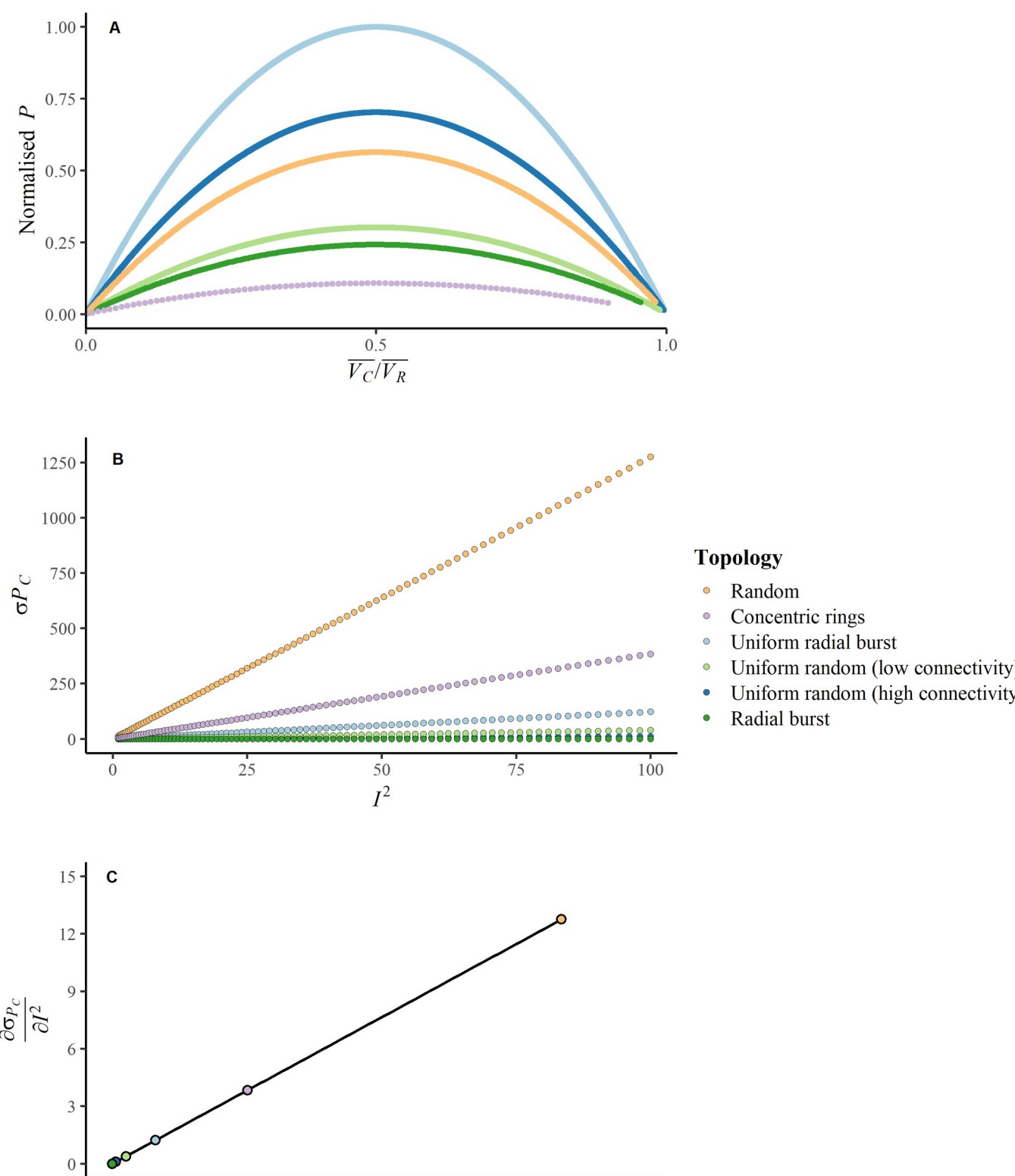

**Fig 2.** For the six example networks, (a) the relationship between total final power ($P$) and the ratio of mean consumer potential to mean resource potential ($\overline{V_C}/\overline{V_R}$), (b) the relationship between the standard deviation of consumer final power ($\sigma_{PC}$) and resource flow squared ($I^2$), and (c) the relationship between the

slope of (b) and the standard deviation of effective resistance ($\sigma R_E$). These illustrate the main equations derived in the presentation of the modelling framework. Here, each coloured point range represents a different network topology over which the simulations were run. The slope of (c) is exactly 1. Units are generalised units of power, potential, and resource flow. A copy of (a) with raw data is included in S1 Fig.

branching pattern around a single resource ('fully branched' network, Fig 4A). In the second, a branching network was artificially evolved from a nearly radial burst pattern, by adding in consecutive levels of non-demand junctions or 'branch points,' and re-calculating the consumption ('evolved branching' networks, Fig 4B). In the 'evolved branching' networks, at each iteration of the evolution, the average link length became shorter, and the network became more similar to a fractal branching structure. This was done to observe how power consumption was affected by changing the architecture to reflect known optimal distribution patterns, without increasing the number of consumers in the network.

In the 'fully branched' network, both the total quantity of power consumption, and inequality of consumer potential and power, were considerably higher than in the other architectures illustrated in Fig 5 (see S1 Table). In contrast, the 'evolved branching' simulations showed lower total power consumption and no inequality present in the final stage of network evolution, as the consumers were all placed equal path distances from the resource, despite being at slightly differing Euclidean distances. This demonstrates that the self-similar branching architecture itself does not lead to inequality, but rather the hierarchical or otherwise heterogeneous distribution of consumers.

In the fully branched network, the underlying hierarchical spatial distribution of consumer nodes and links led to a highly skewed distribution of consumer potentials and final power at

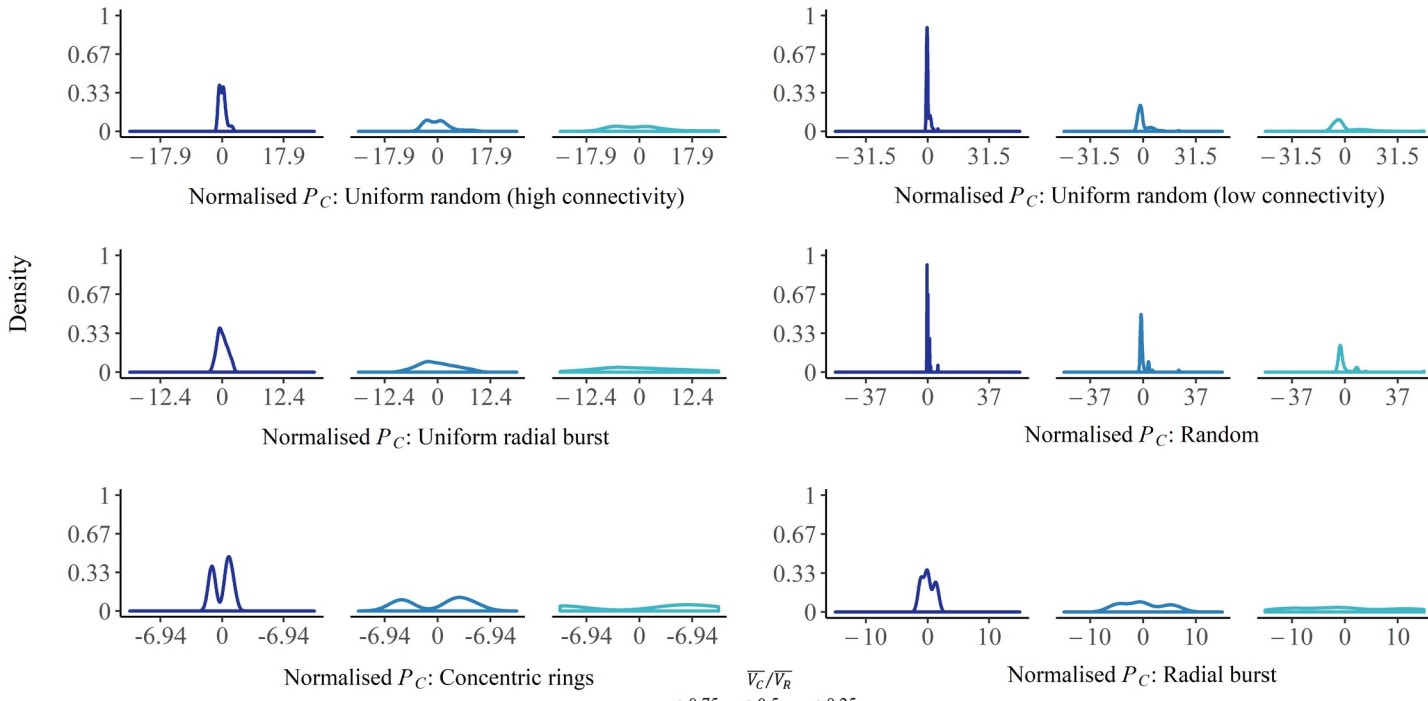

**Fig 3. Density plots of normalised consumer final power (PC) for the six example networks, shown over decreasing ratios of mean consumer to mean resource potential, $\overline{V_C}/\overline{V_R}$.** Each plot shows the density for the normalised consumer final power at $\overline{V_C}/\overline{V_R}$ = 0.75, 0.5, and 0.25, from left to right, as the ratio decreases due to increased resource flow during the simulation. The data were normalised by subtracting the mean consumer power at each ratio level, and dividing by the standard deviation of consumer power at $\overline{V_C}/\overline{V_R}$ = 0.75, such that the width of the first subplot for each network is one standard deviation.

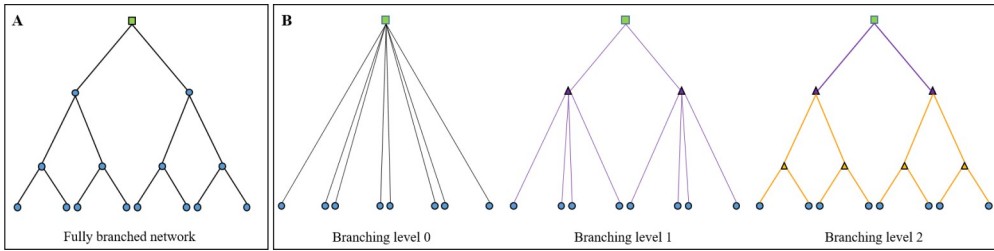

**Fig 4.** (a) A 'fully branched' network, with consumers at each junction, and (b) 'evolved branching' networks illustrating the addition of branch points and links over the course of the simulations. In each network, the green square is the resource, and the blue circles are consumers. In the 'evolved branching' networks, the branch points, represented by triangles, and links of the same colour denote when they were added during the evolution of branching: black links are the original network with no branch points, purple links and branch points are the first level of branching, and gold links and branch points are the second level, which also includes some branch points from previous levels. The network shown here is simplified for illustration purposes: the simulated 'evolved branching' networks contained seven levels of branch points at the final stage of development, and 512 consumers.

network maximum final power, which appears to show power-law properties (Fig 6). While the focus of the work here is on spatial networks, hierarchies can also emerge in relational 'scale-free' networks. These are often represented as hub-and-spoke topologies, with power

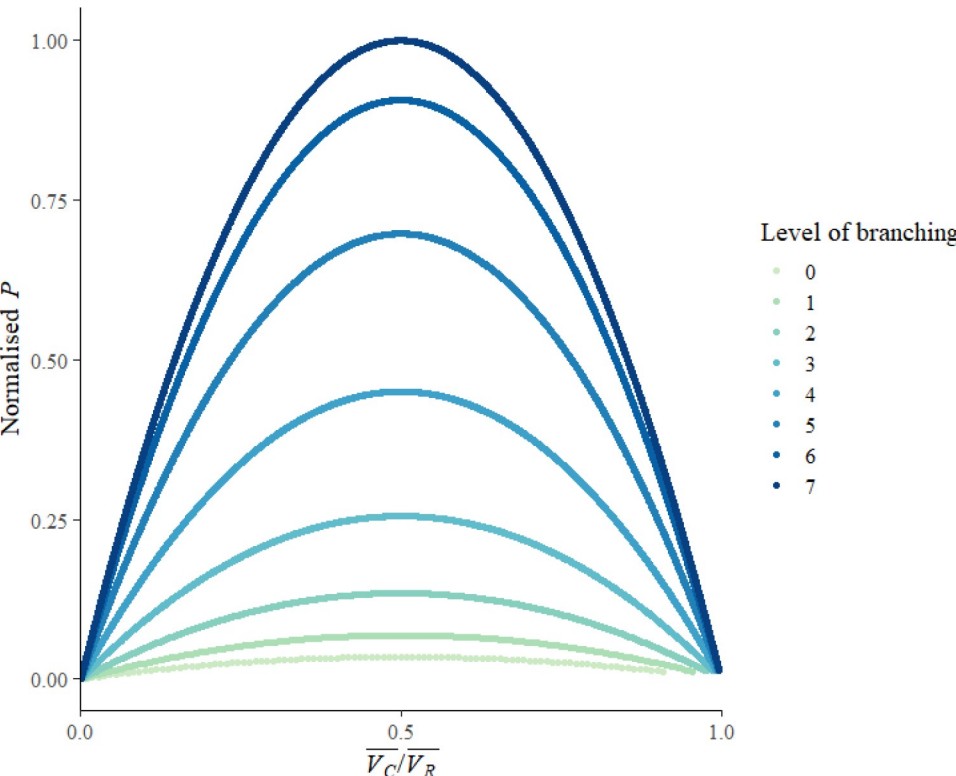

**Fig 5. The total final power consumption (P) against the ratio of mean consumer potential to mean resource potential ($\overline{V_C}/\overline{V_R}$) for each level of the 'evolved branching' networks.** With additional levels of branching, the network became more similar to a fractal branching structure: average link length shortened, and resource flow was concentrated onto fewer, more shared links. Here, each coloured point series represents the trajectory of final power consumption as the network became more branched: Level 0 had no branch points, and Level 7 was a fully self-similar fractal. Relative total final power is the sum of final power consumption at all consumer nodes, normalised by the maximum power achieved by the network, which preserves relative differences. A copy of the figure with raw data is included in S2 Fig.

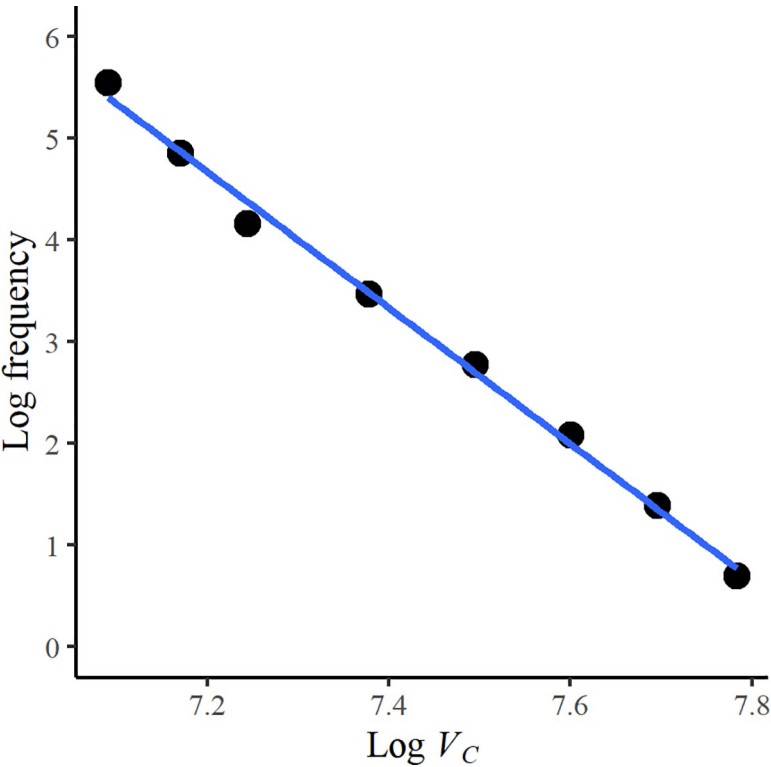

**Fig 6. The frequency distribution of consumer potentials ($V_C$) at maximum network final power for a hierarchically branched network, plotted on log-log axes.** The highly heterogeneous consumer potentials are due to the hierarchical network structure shown in Fig 4A.

law distributions of node degrees. Power law or similarly heavy-tailed distributions in physical systems are typically described as resulting from interactions between interdependent components [36], but the simulations here demonstrate how this distribution can also occur as a result of the spatial organisation of interacting components. It is therefore possible that similar processes give rise to scale-free characteristics both spatially, as in self-similar hierarchical branching, and relationally, as in a power-law distribution of node degrees.

Notably, although self-similar hierarchical branching networks such as the 'fully branched' network can achieve a higher maximum power at the network level, most individual consumers would have higher power if they had direct links to the resource, such as in the radial burst networks. Therefore, branching is still only energetically advantageous to the overall system, and those positioned close to the resource within the network architecture. In addition, these optimally located and connected consumers experience increased final power even after the total network final power begins to decrease (Fig 7), due to the larger frictional losses experienced by the more distant consumers along the bottom level of the network, who have higher effective resistance (Fig 4A). This suggests that hierarchical organisation is only beneficial to the system if the consumers located further from the resource benefit from the overall system operating at a higher maximum power: the more peripheral elements need to gain some of the system-level returns. One example of hierarchical branching as a system-optimal configuration in this way is in the circulatory system of some organisms, where more distant organs and limbs may benefit from the hierarchical organisation of the whole system, even if their individual blood pressure and oxygen levels are lower. Alternatively, if the consumers in more energetically privileged locations exerted enough dominance over the system, the hierarchy could

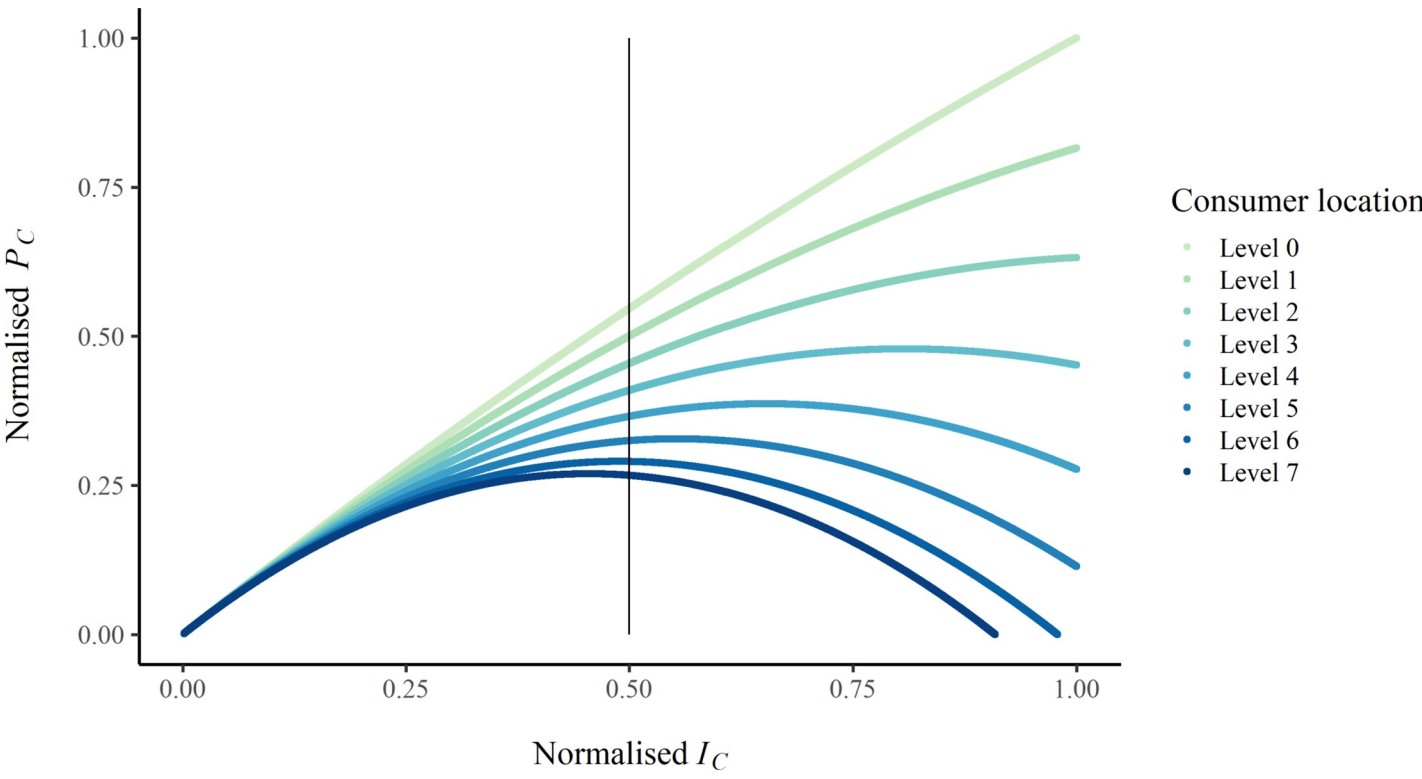

**Fig 7. The relative final power of consumers ($P_C$) at each level of the 'fully branched' network, as related to the relative resource flow to each consumer ($I_C$).** The vertical black line denotes the relative resource flow associated with network-wide maximum final power. Each series represents the relative final power consumption of consumers at that level in the network, where Level 0 is the consumers closest to the resource, and Level 7 are the consumers furthest from the resource in the network. As the resource flow increases across the network, the more distant consumers experience disproportionally greater frictional losses and therefore power losses, while consumers closer to the resource continue to increase in power. Values have been normalised by the maximum consumer final power and maximum consumer resource flow, which preserves relative differences. A copy of the figure with raw data is included in S3 Fig.

be enforced despite being sub-optimal for more distant consumers, and potentially the network as a whole (Fig 7).

## Conclusion

This work has explored the characteristics of complex networks evolving toward maximum power production, and the relationship between the development and dynamics of these networks and the inequality of resource distribution through them. The derived equations and illustrative simulations related the potential, resource flow, power, and resistance across a network of resources and consumers, and illustrated how those relationships changed as the network evolved toward maximum power, through adaptation in network state, architecture, or both. Specifically, it was shown that if the network structure consists of unequal link resistances, resulting from heterogeneity in path distance or connectivity in the network, the inequality of resource distribution will increase as the quantity of resource flow across the network increases. The potential for this architecturally-driven inequality is seen most prominently in hierarchical structures, such as the branching architectures common across in biological, environmental, and human-engineered systems (see e.g. [19,37–39]).

Additionally, this hierarchical branching was shown to only increase the energy transferred through the network at maximum power at the scale of the entire network, and for the consumers located and connected closely to the resources. In contrast, more distant consumers in

these architectures experienced rapid decreases in energy consumption as the resource flow through the network increased, due to higher frictional losses of energy in transport. While prescription is not a focus of the current work, it has illustrated how RADE networks, and specifically hierarchical branching architectures, can be fundamentally linked to the deep inequality experienced by those served by these networks. Explicitly structuring these networks in an attempt to equalise distribution could take the form of co-locating resources and end-users to the greatest extent possible, such as locating solar panels or other forms of renewable energy on homes and businesses [40], or increasing the integration of locally-sourced products into a community's food system [41]. Additional efforts, such as intentionally improving RADE network infrastructure to currently underserved populations of end users [42], could also be a significant step in the right direction. The question remains, however, as to whether even the best efforts at improving equality of distribution can offset the argued thermodynamic trajectory for systems to develop increasing patterns of consumption and dissipation (see e.g. [12]), which appears to be most effectively facilitated by inherently unequal distribution networks.

## Materials and methods

### Required simulation inputs

The simulation code required a Comma-Separated Values (CSV) file to specify parameterisation, including the number of nodes of each type, the size and shape of the spatial topology where they were distributed, whether links were all unit strength or potentially heterogeneous, and the file paths of the CSV files storing the locations of the nodes, or specifying random consumer placement. A complete list of the parameters required, and a description of each, is listed in Table 1.

**Table 1. Modified load flow methodology input parameters and description.**

| Parameter name | Description |
| --- | --- |
| topology | The name of the shape in or on which the nodes are distributed. Values: SPHERE (nodes located within a sphere of a given radius), SPHERE_SURFACE (nodes located on the surface of a sphere a given radius),PLANE (nodes located on the surface of a plane). |
| pNoConnection | The probability of two nodes not connecting, in a network with random links. |
| noConnection | The placeholder value in the connections matrix for non-connected nodes. |
| resourcesFile | The file path of the CSV file storing the coordinate locations and potentials of the resources. |
| planeMaxCoords | The maximum coordinates of the plane, stored as a pair of values separated with a semi-colon (e.g. 100;100). |
| sphereR | The radius of the sphere, or sphere surface. |
| nBranchPoints | The number of branch points. |
| nConsumers | The number of consumers. |
| useStrength | Whether or not to use link strength in calculating the resistance between nodes. Values: TRUE/FALSE. |
| strengthExponent | The exponent to which the link strength, if used, should be raised. |
| manualNetwork | Whether to read in a pre-specified connections matrix or generate the links randomly. Values: TRUE/FALSE. |
| randomConsumers | Whether to distribute the consumers randomly in the topology or use specified locations. Values: TRUE/FALSE. |
| consumersFile | The file path of the CSV file storing the coordinate locations of the consumers (if not random). |
| matrixFile | The file path of the CSV file storing the connections matrix, if a pre-specified one is used. |
| branchPointsFile | The file path of the CSV file storing the coordinate locations of the branch points (if used). |
| outputCSV | The file path to the CSV file where the output of the code run is stored. Includes the resource flow specification per consumer, the power and potential at each resource and consumer, and the total link length of the network. |

The topologies simulated here included planes, spheres, and sphere surfaces. Planes and spheres can be classed as two- and three-dimensional spaces, respectively, while sphere surfaces are of a more ambiguous dimension (see e.g. [1]). The exploration of these three relevant topologies, commonly used to represent idealised spaces in physical systems, allowed identification of any effect on power consumption or resource distribution due to dimensionality. In these networks, the size of the topology, measured in generalised units as the radius of the sphere or sphere surface, or one side of the square plane, was determined by the number of nodes of each type,

$$Size = \sqrt{10nC * 100nR},\qquad(6)$$

where $nC$ is the number of consumers, and $nR$ is the number of resources. This was chosen as it allowed for meaningfully large distances between nodes in networks with multiple consumer and resource nodes. The branched networks had set lengths for each link, such that topology size was not a factor.

The relationship between spatial size and power distribution and consumption was not directly explored, such as by spreading the same network architecture across a larger area, but the linearity of the equation for resistance with unit-strength links suggests that inequality in consumer potential would increase linearly, and power consumption would decrease linearly, with increases in topological size. Similarly, the resource potentials were chosen to provide a clear visualisation of the maximum power 'curve' (Fig 2A), but a range was not explored, as increasing or decreasing the resource potential(s) would simply linearly increase or decrease the consumer potentials (see Eq 1).

In all simulations with random and radial burst topologies, link strength was set to 1. In the branching simulations, it was set to be proportional to the resource flow, squared, to offset the increased frictional losses from higher resource flow along shared links. Specifically, by re-arranging Eq 1, the potential gradient along a link can be calculated as a product of resource flow $I$ and link resistance $R$. Recall that power loss along a link $P_L$ is a product of this potential gradient and resource flow along it (Eq 3), which when combined with Eq 1 gives

$$P_L = \frac{I^2 L}{S}.\qquad(7)$$

Since losses are proportional to the resource flow squared, it rapidly dominates the energy losses. Therefore, as branching networks combine resource flows onto shared branches, they experience higher flow-driven losses on those shared links, despite having lower total network resistance, due to the shared links shortening the total path length around the network. It follows that, for the branching to be energetically advantageous, the link strength must be a function of resource flow, $S = f\{I\}$. If $P_L \propto I^2$, then $S \propto I^2$, resulting in the power loss becoming a function exclusively of link length (Eq 7). This allows the advantages of shorter total link length in a branching network to be realised.

## Simulation code operation

An overview of the simulation code is shown in Fig 8. After the program read in the specified parameters above, it created a customised data structure to store the node locations, resource potentials, and connections matrix. If the consumer locations were random, the program placed each consumer in space by drawing each coordinate from a uniform distribution bounded by the maximum topology coordinates supplied. If the links were random, the program put a link between each node, except for resources, with the probability of 1 –pNoConnection parameter described above (Table 1).

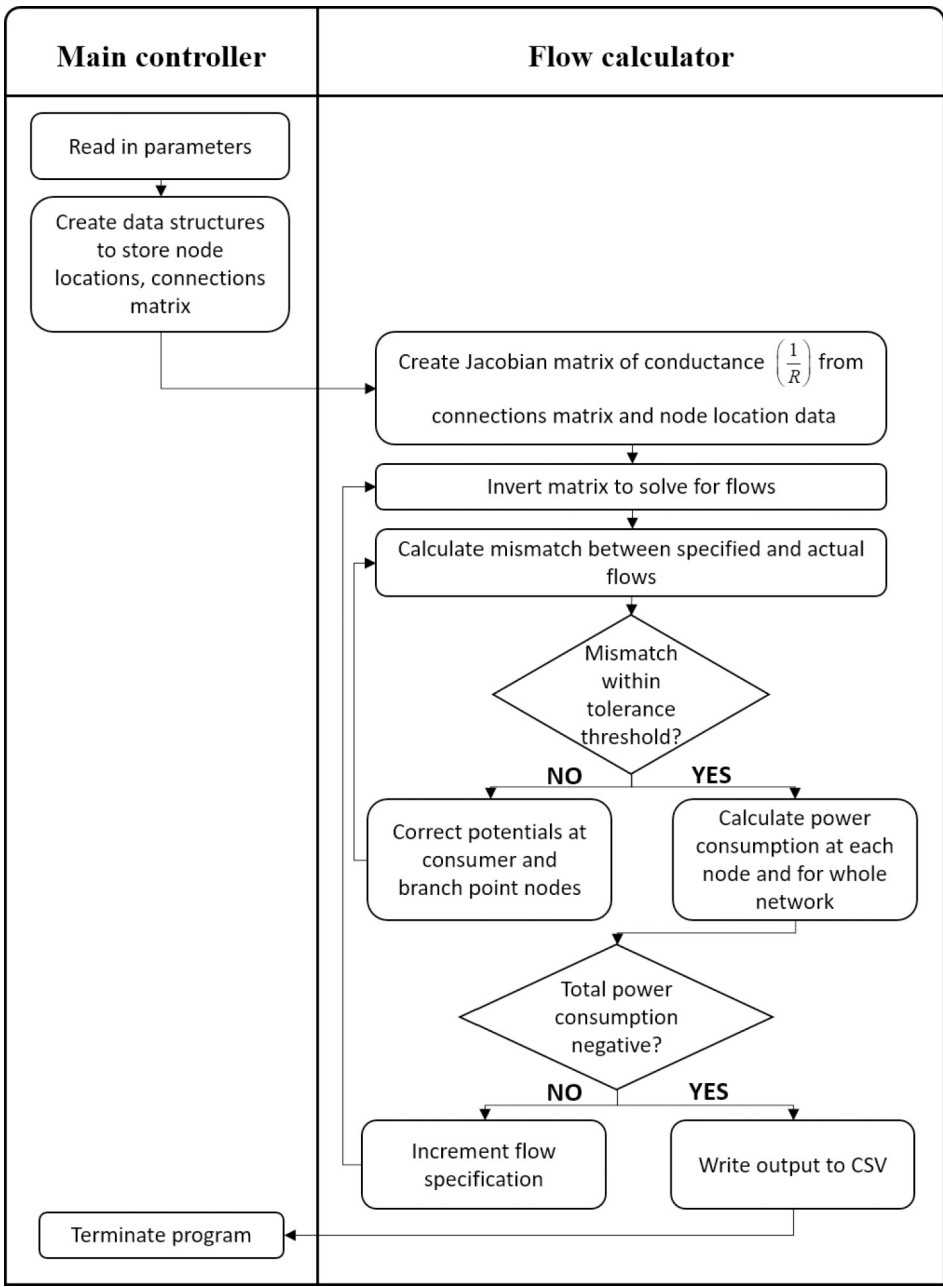

**Fig 8. Code flow diagram for flow calculator program.** The main controller of the program reads in the parameters and creates the network, and eventually terminates the program when complete, while the main calculations of the program are based on an iterative matrix inversion process in the flow calculator class.

To calculate the resource flows and power consumption of the network, the program constructed a Jacobian matrix representing the conductance of the network, or the inverse of the resistance, using the connections matrix. This was inverted to solve for the mismatch between specified and received resource flow at each consumer node, based on the consumer potentials. These were determined by the load flow equations using a matrix form of Eq 1.

The potentials at the consumers, and branch points if used, were then adjusted to counter the mismatch. The matrix inversion and mismatch calculations were repeated until the

mismatches were within the specified error threshold of 0.001. After convergence, the total power consumption of the network $P_{Network}$ was calculated as the sum of the power consumption at each consumer $P_{C_i}$, which was the product of potential $V_{C_i}$ and resource flow $I_{C_i}$ (Eq 2):

$$P_{Network} = \sum_{i=1}^{nConsumers} P_C. \tag{8}$$

Initially, the total power consumption was calculated for 1 unit of resource flow arriving at each consumer. With each iteration, the specification was incremented by 0.1 unit, and the resource flows were re-calculated. This was repeated until either 1000 units of resource flow was arriving at each consumer, or the power consumption of the network was negative, due to the inverse relationship between consumer potential and resource flow (Eq 1). In the evolving branching simulations, an additional level of branch points was added, and the links between nodes re-arranged, after the iterations had completed for a given network, until there were 7 levels of branch points between the resource and consumers (Fig 4B).

## Simulation outputs

The program output was a single CSV file, with the potential at each resource and consumer, and the power production and consumption of each resource and consumer, respectively, at each resource flow specification tested. It also included the total link length of the network, which does not change over the duration of the simulations.

## Code availability statement and languages used

A complete copy of the code, along with usage instructions, a sample parameter file, and sample resource, consumer, branch point, and matrix CSVs, is available upon request. The code is written in Java Version 8. All figures and analyses were generated using R, including the base R package version 3.6.1 [43], ggplot2 [44], and the rgl package [45].

## Supporting information

**S1 Text. Maximum power derivation.**
(DOCX)

**S1 Table.** Parameterisation and power consumption details of networks simulated: a) branched networks, and b) random and radial burst networks. The slope of the linear relationship between $\sigma_{P_C}$ and $I^2$ shown in Fig 1 was calculated for each plot using least-squares regression, and compared to the value of $\sigma_{R_E}$ calculated using Eq 5 and the consumer and resource potentials for each network, shown in the table here. The least-squares regression estimate of $\sigma_{R_E}$ is shown in brackets below the original estimate using consumer and resource potentials, for the networks plotted.
(DOCX)

**S1 Fig. Non-normalised version of Fig 2A, showing the relationship between total final power (P) and the ratio of mean consumer potential to mean resource potential $(\overline{V_C}/\overline{V_R})$ for six example networks.** Each coloured point range represents a different network topology over which the simulations were run. The units are generalised units of power, rather than units only applicable to a specific type or types of resource distribution network.
(TIF)

**S2 Fig. Non-normalised version of Fig 5, showing total final power consumption (P) against the ratio of mean consumer potential to mean resource potential $(\overline{V_C}/\overline{V_R})$, for the**

**'evolved branching' networks.** The units are generalised units of power, rather than units only applicable to a specific type or types of resource distribution network.
(TIF)

**S3 Fig. Non-normalised version of Fig 7, showing final power of consumers ($P_C$) at each level of the 'fully branched' network, as related to the resource flow to each consumer ($I_C$).** The units are generalised units of power and resource flow, rather than units only applicable to a specific type or types of resource distribution network.
(TIF)

## Author Contributions

**Conceptualization:** Natalie Davis, Andrew Jarvis, J. Gareth Polhill.

**Data curation:** Natalie Davis.

**Formal analysis:** Natalie Davis.

**Investigation:** Natalie Davis.

**Methodology:** Natalie Davis.

**Project administration:** Natalie Davis.

**Software:** Natalie Davis.

**Supervision:** Andrew Jarvis, M. J. Aitkenhead, J. Gareth Polhill.

**Visualization:** Natalie Davis.

**Writing – original draft:** Natalie Davis.

**Writing – review & editing:** Natalie Davis, Andrew Jarvis, M. J. Aitkenhead, J. Gareth Polhill.

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
