## [Decision Letter · Decision Letter 0]

17 Dec 2019

PONE-D-19-29844

Trajectories toward maximum power and inequality in resource distribution networks

PLOS ONE

Dear Ms. Davis,

Thank you for submitting your manuscript to PLOS ONE. After careful consideration, we feel that it has merit but does not fully meet PLOS ONE’s publication criteria as it currently stands. Therefore, we invite you to submit a revised version of the manuscript that addresses the points raised during the review process.

Your manuscript has been seen by two reviewers.  Both indicated that your work had merit, but the exposition of your ideas and results needed substantial improvement to make it more accessible to the diverse readership of PLoS ONE.  Comments from both reviewers, especially Reviewer 2, provide specific pointers for potential improvements.  Please go through both reviewers' comments carefully and address them properly in your revision.

We would appreciate receiving your revised manuscript by Jan 31 2020 11:59PM. To enhance the reproducibility of your results, we recommend that if applicable you deposit your laboratory protocols in protocols.io, where a protocol can be assigned its own identifier (DOI) such that it can be cited independently in the future. For instructions see: http://journals.plos.org/plosone/s/submission-guidelines#loc-laboratory-protocols

We look forward to receiving your revised manuscript.

Kind regards,

Rachata Muneepeerakul

Academic Editor

PLOS ONE

Journal Requirements:

Reviewers' comments:

Reviewer's Responses to Questions

**Comments to the Author**

1. Is the manuscript technically sound, and do the data support the conclusions?

Reviewer #1: Yes

Reviewer #2: Partly

2. Has the statistical analysis been performed appropriately and rigorously? 

Reviewer #1: Yes

Reviewer #2: N/A

3. Have the authors made all data underlying the findings in their manuscript fully available?

Reviewer #1: Yes

Reviewer #2: Yes

4. Is the manuscript presented in an intelligible fashion and written in standard English?

Reviewer #1: Yes

Reviewer #2: Yes

5. Review Comments to the Author

Reviewer #1: In this paper the authors address an optimization problem, in which the goal is to model and analyze the power consumption in spatially explicit resource distribution networks. Using an analogy with electrical current and Ohm's law the authors are able to describe the overall behavior of the system. They also analyzed how a hierarchical structure affects the power consumption. I found that the results are interesting and the manuscript is scientifically sound. I recommend this paper to be published provided the authors take care of the following remarks in a satisfactory way.

- The conclusion is a section meant to summarize and re-state the main findings of the manuscript. In this case, the text after line 377 motivates the importance of the work done, but I believe those arguments belong to the introductory section. Conclusion should be re written to emphasize the new results.

- A very common example of heterogeneous complex network that is ubiquitous across nature are scale free networks. Even though the authors analyzed several network topologies, completely overlooked this particular category of graphs. Is there a specific reason for that?

- Axis on Fig. 6 should include the symbols used for those quantities, to be in agreement with the text. For example ``Log consumer potentials'' I assume should be ``$\\log(\\overline{V_c}) $''. The same could be applied to some of the plots on Fig. 2, Fig. 3, and Fig. 7.

Reviewer #2: I apologize for the delay -- this has been a busy time in the semester!

This paper explores the relation between the topology of a RADE network and its heterogeneity, in terms of both the spatial distribution of the underlying resource density and of end-user service, the latter being interpreted as a measure of heterogeneity. The first part of the paper develops an electical analog to explore the relation between resource heterogeneity (measured as the standard deviation of potential) and inequality (measured as the standard deviation of end-user power delivery) for given network topologies. The second part of the paper releases the constraint of network topology and allows it to self organize by branching out in a hierarchical system so as to minimize power transmission losses. Results appear to describe an interesting tension between node-optimal and system optimal configurations: the former emerges when consumers are directly connected to the resource (e.g., radial burst), the latter emerges in hierarchical topologies. They use this insight to explain the emergence of branching in organisms (where system-optimal configuration prevails) and in networks with strong assymetries of power (i.e. where energetically-privileged consumers dominate).

In substance, the paper makes an elegant and substantial contribution to the field and I would be very much looking forward to it being published in PLOS ONE. I believe, however, that the impact of the manuscript could be improved by a better exposition, which would make it more accessible to a wider audiience. Specifically:

1. It would be beneficial to specify clearly and early enough what a RADE network is and is not. Being not familiar with the concept, it took me a while to understand that the network relates two types of nodes: “resources” and “consumers”. It is still not clear to me whether a node can be both a resource and a consumer at the same time. Also, it took me a while to understand that the “potential” that generates the flow cannot be “depleted” in the sense that V(x,y) is not affected by i. Do your results hold in a (arguably more realistic) resource network, where the potential can be depleted?

2. In terms of what a RADE network is not, it may be beneficial to exclude networks of agents, where the particles that move do not follow an exogenous gradient but rather “decide” where to go based on an individual level optimization. A number of folks that think about networks (including myself), think about them in the context of trade and migration, both of which likely behave quite differently from what you describe.

3. In general, it would be helfpul to add one sentence to each caption with the figure’s main point. Figures 6 and 7 are particularly opaque and need more unpacking, both in the caption and in the text. I am having a lot of troubles figuring out how to interpret them.

4. The notions of heterogeneity and inequality appear to get diluted as you move towards self organizing networks. Do you still measure them as standard deviations of U and P? If so, I am wondering if these metrics are still appropriate representation of heterogeneity and inequality in a complex network, where the relation between topology and heterogeneity matters. For instance, does a variation of P across a “stem” link have an equivalent effect on network-level inequality as the same variation across a “branch” link? These matters are central to the topic of the paper (at least I would think) and merit further discussion.

L165. An example/case to illustrate your point on the pitfalls of using a GINI coefficient would be useful here.

L296. A figure showing the difference between the two compared networks and what you mean by “branching level” would be helpful here. Perhaps display a few stages of the evolution between a radial blast and a branched network?

L344-356 appear very important, but I had to read them several times to understand what you mean (and I’m not even sure I did understand —see my attempt at summarizing them in the first paragraph of this review). I think they merit to be reformulated and clarified.

L377. “Inequality of distribution is not intrinsiqually inequitable”. Not sure what you mean. How do you define inequitable. Please be more specific.

6. PLOS authors have the option to publish the peer review history of their article (what does this mean?). If published, this will include your full peer review and any attached files.

Reviewer #1: No

Reviewer #2: No

---

## [Author Response · Author response to Decision Letter 0]

29 Jan 2020

Dear Dr. Muneepeerakul,

Thank you for your reading and review of our manuscript, and the opportunity to revise and resubmit. The reviewers’ constructive comments were very helpful in identifying areas to clarify, and we respond to each below, referencing line numbers of changed text in the unmarked revised manuscript version.

In response to reviewer comments, we reformulated the conclusion to focus more on the new findings presented in the text, added some text discussing the relationship between relational and spatial networks to improve the exposition to a wider audience, and clarified our definitions of terminology throughout. In addition, we moved two paragraphs to the methods section and removed a related illustrative figure, as these provided interesting but non-essential background material, which may have distracted from the main story presented. We have also incorporated suggested changes to formatting, such as figure axes.

Thank you for your consideration of our resubmission, and we look forward to your response.

Sincerely,

Natalie Davis (corresponding author)

ndavis.research@gmail.com

+44 7434 547051

Andrew Jarvis

M.J. Aitkenhead

J. Gareth Polhill

Reviewer #1: In this paper the authors address an optimization problem, in which the goal is to model and analyze the power consumption in spatially explicit resource distribution networks. Using an analogy with electrical current and Ohm's law the authors are able to describe the overall behavior of the system. They also analyzed how a hierarchical structure affects the power consumption. I found that the results are interesting and the manuscript is scientifically sound. I recommend this paper to be published provided the authors take care of the following remarks in a satisfactory way.

- The conclusion is a section meant to summarize and re-state the main findings of the manuscript. In this case, the text after line 377 motivates the importance of the work done, but I believe those arguments belong to the introductory section. Conclusion should be re written to emphasize the new results.

We included that information in the conclusion rather than introduction as we thought it made most sense when presented in light of the results, which showed how inequality could develop within networks. In addition, we provided further discussion on how our findings could be applied, although the paper was not structured as a prescriptive piece. In consideration of this comment, however, we have revised the conclusion (L 366-392) to include a summary of the key points of the manuscript and removed the discussion of inequality as not necessarily negative, to clarify the story. 

- A very common example of heterogeneous complex network that is ubiquitous across nature are scale free networks. Even though the authors analyzed several network topologies, completely overlooked this particular category of graphs. Is there a specific reason for that?

This is an excellent question, and raises an interesting point about the mapping between spatial and relational networks. Traditionally, scale-free networks are defined as those where the node degree distribution follows a power law. These are argued to be both ubiquitous (Barabási & Albert, 1999) and rare (Broido & Clauset, 2019) in natural and human-engineered networks, depending on how the authors define a power law (Holme, 2019). We did not investigate scale-free networks explicitly as their definition is not inherently spatial, which was our focus: many of the networks suggested to exhibit scale-free features are social or interactional networks (see e.g. Pastor-Satorras & Vespignani, 2001; Jones & Handcock, 2003). The spatial variant of scale-free networks are those which are replicates of themselves across scales (West et al., 1997), e.g. fractal branching structures, which were investigated. 

However, we agree that relational networks are crucial, and would argue that all networks are necessarily both interactional and spatial. It is quite possible that the process that gives way to spatially scale-free networks, e.g. fractal branching structures, is related to those that create relationally scale-free networks, e.g. those with a power law degree distribution. Future work exploring the mapping between these would be fascinating, and would likely include a metric defined similarly to the effective resistance (RE) that we derived, as this incorporates both structural and connectivity effects. We have added some text to the section describing the modelling framework (L 204-207) and the section analysing simulations of branched networks (L 323-332) exploring this.

Additionally, we emphasise that the results we present hold for all networks, regardless of topology or degree distribution. Although we did not choose to explicitly discuss relationally scale-free networks, the same results would apply. See below for plots from a simple hub-and-spoke topology that was scale-free in the node degree distribution, which had 90 consumers and 10 resource nodes, located at the ‘hubs’ of the network. When we ran the simulations over it, the same major results found across other networks held: maximum power occurred when the potential at the average consumer was 50 % of the potential at the average resource, and the standard deviation of consumer potential increased linearly with the square of the resource flow.

- Axis on Fig. 6 should include the symbols used for those quantities, to be in agreement with the text. For example ``Log consumer potentials'' I assume should be ``$\\log(\\overline{V_c}) $''. The same could be applied to some of the plots on Fig. 2, Fig. 3, and Fig. 7.

We have changed the axes in all figures to use the symbols referred to by the text and added definitions of those symbols to the figure captions for clarity.

Reviewer #2: I apologize for the delay -- this has been a busy time in the semester!

This paper explores the relation between the topology of a RADE network and its heterogeneity, in terms of both the spatial distribution of the underlying resource density and of end-user service, the latter being interpreted as a measure of heterogeneity. The first part of the paper develops an electrical analog to explore the relation between resource heterogeneity (measured as the standard deviation of potential) and inequality (measured as the standard deviation of end-user power delivery) for given network topologies. The second part of the paper releases the constraint of network topology and allows it to self organize by branching out in a hierarchical system so as to minimize power transmission losses. Results appear to describe an interesting tension between node-optimal and system optimal configurations: the former emerges when consumers are directly connected to the resource (e.g., radial burst), the latter emerges in hierarchical topologies. They use this insight to explain the emergence of branching in organisms (where system-optimal configuration prevails) and in networks with strong asymmetries of power (i.e. where energetically-privileged consumers dominate).

In substance, the paper makes an elegant and substantial contribution to the field and I would be very much looking forward to it being published in PLOS ONE. I believe, however, that the impact of the manuscript could be improved by a better exposition, which would make it more accessible to a wider audience. Specifically:

1. It would be beneficial to specify clearly and early enough what a RADE network is and is not. Being not familiar with the concept, it took me a while to understand that the network relates two types of nodes: “resources” and “consumers”. It is still not clear to me whether a node can be both a resource and a consumer at the same time. Also, it took me a while to understand that the “potential” that generates the flow cannot be “depleted” in the sense that V(x,y) is not affected by i. Do your results hold in a (arguably more realistic) resource network, where the potential can be depleted?

Thank you, this is an excellent suggestion. We have added L 24-27 at the beginning of the introduction to define RADE networks more clearly and clarify what is in scope for the work we present. Additionally, we added L 211-215 to the brief section outlining the simulations, to clarify that a consumer node can pass on excess resource flow to further ‘downstream’ consumers, but cannot generate additional resource flow, i.e. cannot be a resource as well. We note that RADE networks in reality are nested within one another, such that one network’s consumer is another network’s resource, but for clarity this is not represented in our simulations.

With regards to your point about potential depletion, we chose to investigate the evolution toward maximum power in RADE networks with constant resource potential, similarly to time-averaged behaviour of renewable or regenerating resources, or development of a network over a timescale in which the resource does not deplete. While the evolution of networks with finite resources is a fascinating topic, and one that we are working on another paper to investigate, we do not cover it explicitly here. However, we have added L 263-268 to discuss our prediction of how our findings would apply in that situation: namely, that the networks would become more unequal, more quickly, as the more distant and/or less directly connected consumers would experience the effects of the decreasing resource potential more quickly, as they already operate at lower potentials due to frictional losses of energy in transport to them.

2. In terms of what a RADE network is not, it may be beneficial to exclude networks of agents, where the particles that move do not follow an exogenous gradient but rather “decide” where to go based on an individual level optimization. A number of folks that think about networks (including myself), think about them in the context of trade and migration, both of which likely behave quite differently from what you describe.

The amount of distinction that can be drawn between gradient-based networks and decision-based networks is a very interesting point. We would argue that even in networks of agents, each seeking an individual optimum, the decision space is restricted by the amount of free energy with which agents can move themselves, their resources, or both. This free energy supply is governed by the existing network structure and energy flows, as determined by resource and network characteristics. In your work with trade networks, this would be illustrated by agents only being able to make trade decisions based on what they already have (or possibly their political power, which is arguably based on consumption or trade of past resources). While our work is not explicitly focussed on networks of moving agents, we argue that many of the same principles apply. We have attempted to draw additional connections between spatial and interactional networks in our clarification of L 323-332 as well.

3. In general, it would be helpful to add one sentence to each caption with the figure’s main point. Figures 6 and 7 are particularly opaque and need more unpacking, both in the caption and in the text. I am having a lot of troubles figuring out how to interpret them.

We added L 242-243 to the caption for Fig 2, added L 249-251 to the caption for Fig 3, added and clarified L 315-319 to the caption for Fig 5, L 334-335 to the caption of Fig 6, and L 355-360 to the caption for Fig 7. We also corrected the legend titles for Figs 5 and 7. Additionally, the changes to the main text in L 323-332 should help further clarify Fig 6, and L 337-352 should help further clarify Fig 7.

4. The notions of heterogeneity and inequality appear to get diluted as you move towards self organizing networks. Do you still measure them as standard deviations of U and P? If so, I am wondering if these metrics are still appropriate representation of heterogeneity and inequality in a complex network, where the relation between topology and heterogeneity matters. For instance, does a variation of P across a “stem” link have an equivalent effect on network-level inequality as the same variation across a “branch” link? These matters are central to the topic of the paper (at least I would think) and merit further discussion.

We are not entirely clear what this question is asking. In the paper, inequality is measured as the standard deviation of consumer power, for all networks. Furthermore, our focus throughout is on the relationship between topology and heterogeneity, hence the derivation of RE (Eq. 5) and discussion of how the hierarchy distribution of consumers, such as in some branching networks, causes increased inequality of distribution (see L 310-311 for a clarification here). We would argue that heterogeneity and inequality are in fact amplified by self-organisation in complex networks, such as in the global economy. In our reworking of L 337-352 (in response to your point below), we have attempted to clarify this further. We have also moved the sections about link strength being proportional to resource flow (previously L 268-294) to the Methods (now L 424-437), and removed what was previously Figure 4, as these sections were very interesting but possibly distracting from the focus of the manuscript on resource flows and inequality. By moving a substantial section of this information to the methods and referring to it in the main text as necessary (e.g. L 284), the main story flows more clearly from the sections introducing branching and the simulations, to the discussion of how hierarchy, seen here in the branching networks, affects inequality.

The inequality of distribution within a network is not governed by any one link, however, but rather the distribution of links and node interconnection, leading to our use of RE and its standard deviation to capture these joint effects. In this context, the standard deviation is still appropriate, given that it is a universal measure of dispersion independent of the underlying distribution, since we are not assuming anything about the proportion of observations within a given number of standard deviations from the mean. We have added L 204-207 to elucidate further the applications of the effective resistance.

L165. An example/case to illustrate your point on the pitfalls of using a GINI coefficient would be useful here.

We added L 166-169 briefly showing how increasing each number in a distribution by the same factor increases the absolute inequality (standard deviation and range) but does not increase the Gini coefficient, or relative inequality, of the distribution.

L296. A figure showing the difference between the two compared networks and what you mean by “branching level” would be helpful here. Perhaps display a few stages of the evolution between a radial blast and a branched network?

We moved Figure S3 into the main text (now Figure 4), and broke it into three separate networks to illustrate the evolution of the network with the addition of links and branch points more clearly. We also added Part A to the figure to illustrate the ‘fully branched’ network (as compared to the ‘branching levels’ networks). Finally, we renamed ‘branching levels’ networks to ‘evolved branching’ throughout as we thought this more clearly described the simulations.

L344-356 appear very important, but I had to read them several times to understand what you mean (and I’m not even sure I did understand —see my attempt at summarizing them in the first paragraph of this review). I think they merit to be reformulated and clarified.

Your summary in the first paragraph of the review was very accurate, and captured well what we were trying to express! Although this suggests the paragraph does convey what we intended, we have changed what is now L 337-352 by adding some additional explanation and clarifying some of the terminology and phrasing, to connect more clearly with the concepts and results presented previously. We used some of the terminology from your summary, such as ‘system-optimal configuration,’ in the clarification: that was a very accessible way of making the distinction. 

L377. “Inequality of distribution is not intrinsically inequitable”. Not sure what you mean. How do you define inequitable? Please be more specific.

We removed this sentence during the restructuring of the conclusion, however, by inequitable, we meant a situation in which individuals did not have their needs met. Inequality of distribution in the example systems we listed, such as a body, is to be expected: the liver and stomach require more blood flow than the fingers and toes, so they should receive more blood flow. Inequality of distribution in a network of consumers with similar requirements, however, such as distribution of food or energy resources in human communities, could be considered inequitable if some consumers did not receive what they needed from the network.

---

## [Decision Letter · Decision Letter 1]

19 Feb 2020

Trajectories toward maximum power and inequality in resource distribution networks

PONE-D-19-29844R1

Dear Dr. Davis,

We are pleased to inform you that your manuscript has been judged scientifically suitable for publication and will be formally accepted for publication once it complies with all outstanding technical requirements.

With kind regards,

Rachata Muneepeerakul

Academic Editor

PLOS ONE

Additional Editor Comments (optional):

Reviewers' comments:

Reviewer's Responses to Questions

**Comments to the Author**

1. If the authors have adequately addressed your comments raised in a previous round of review and you feel that this manuscript is now acceptable for publication, you may indicate that here to bypass the “Comments to the Author” section, enter your conflict of interest statement in the “Confidential to Editor” section, and submit your "Accept" recommendation.

Reviewer #1: All comments have been addressed

Reviewer #2: All comments have been addressed

2. Is the manuscript technically sound, and do the data support the conclusions?

Reviewer #1: Yes

Reviewer #2: Yes

3. Has the statistical analysis been performed appropriately and rigorously? 

Reviewer #1: Yes

Reviewer #2: N/A

4. Have the authors made all data underlying the findings in their manuscript fully available?

Reviewer #1: Yes

Reviewer #2: Yes

5. Is the manuscript presented in an intelligible fashion and written in standard English?

Reviewer #1: Yes

Reviewer #2: Yes

6. Review Comments to the Author

Reviewer #1: The authors addressed all my concerns in their response. I recommend this paper to be accepted for publication.

Reviewer #2: I am satisfied by the way the authors have addressed my comments from the previous submission and I am looking forward to seeing this manuscript appear in plos one!

7. PLOS authors have the option to publish the peer review history of their article (what does this mean?). If published, this will include your full peer review and any attached files.

Reviewer #1: No

Reviewer #2: No

---

## [Editor Report · Acceptance letter]

21 Feb 2020

PONE-D-19-29844R1 

Trajectories toward maximum power and inequality in resource distribution networks 

Dear Dr. Davis:

I am pleased to inform you that your manuscript has been deemed suitable for publication in PLOS ONE. Congratulations! Your manuscript is now with our production department. 

With kind regards,

on behalf of

Dr Rachata Muneepeerakul 

Academic Editor

PLOS ONE